# Prioritizing Radiation and Targeted Systemic Therapies in Patients with Resected Brain Metastases from Lung Cancer Primaries with Targetable Mutations: A Report from a Multi-Site Single Institution

**DOI:** 10.3390/cancers16193270

**Published:** 2024-09-26

**Authors:** Yen-Ruh Wuu, Mostafa Kokabee, Bin Gui, Simon Lee, Jacob Stone, Jessie Karten, Randy S. D’Amico, Morana Vojnic, A. Gabriella Wernicke

**Affiliations:** 1Department of Radiation Medicine, Northwell, New Hyde Park, NY 11042-1069, USA; ywuu@northwell.edu (Y.-R.W.); bgui@northwell.edu (B.G.); slee105@pride.hofstra.edu (S.L.); jstone16@pride.hofstra.edu (J.S.); jkarten1@northwell.edu (J.K.); 2Department of Pathology, Lenox Hill Hospital, Northwell, New York, NY 10075-1850, USA; mkokabee@northwell.edu; 3Department of Neurological Surgery, Lenox Hill Hospital, Northwell, New York, NY 10075-1850, USA; rdamico8@northwell.edu; 4Zucker School of Medicine at Hofstra, Hempstead, NY 11549-1000, USA; mvojnic@northwell.edu; 5Department of Medical Oncology, Lenox Hill Hospital, Northwell, New York, NY 10075-1850, USA

**Keywords:** radiation, SRS, chemotherapy, immunotherapy, molecular profiling, brain metastasis, lung cancer

## Abstract

**Simple Summary:**

Patients with brain metastases from non-small cell lung cancer are often managed by various treatment approaches including surgery, chemotherapy/immunotherapy, and radiation therapy. While patients with a good performance status and a limited burden of disease are often candidates for surgical resection, the selection of the optimal adjuvant treatment paradigm, in light of the development of novel treatments for tumors with actionable mutations, remains a challenge. In this study, we sought to examine the benefit of radiation therapy, systemic therapy, or a combination of treatments following surgery with respect to in-brain progression-free survival and overall survival. Our data demonstrate that a combination of radiation therapy and systemic therapy has a benefit in improving progression-free survival in patients with non-small cell lung cancer who have developed brain metastases.

**Abstract:**

**Background/Objectives**: Brain metastases (BrMs) are a common complication of non-small cell lung cancer (NSCLC), present in up to 50% of patients. While the treatment of BrMs requires a multidisciplinary approach with surgery, radiotherapy (RT), and systemic therapy, the advances in molecular sequencing have improved outcomes in patients with targetable mutations. With a push towards the molecular characterization of cancers, we evaluated the outcomes by treatment modality at our institution with respect to prioritizing RT and targeted therapies. **Methods**: We identified the patients with NSCLC BrMs treated with surgical resection. The primary endpoints were in-brain freedom from progression (FFP) and overall survival (OS). The secondary endpoint included index lesion recurrence. The tumor molecular profiles were reviewed. The outcomes were evaluated by treatment modality: surgery followed by adjuvant RT and/or adjuvant systemic therapy. **Results**: In total, 155/272 (57%) patients who received adjuvant therapy with adequate follow-up were included in this analysis. The patients treated with combination therapy vs. monotherapy had a median FFP time of 10.72 months vs. 5.38 months, respectively (*p* = 0.072). The patients of Hispanic/Latino vs. non-Hispanic/Latino descent had a statistically significant worse OS of 12.75 months vs. 53.15 months, respectively (*p* = 0.015). The patients who received multimodality therapy had a trend towards a reduction in index lesion recurrences (χ^2^ test, *p* = 0.063) with a statistically significant improvement in the patients receiving immunotherapy (χ^2^ test, *p* = 0.0018). **Conclusions**: We found that systemic therapy combined with RT may have an increasing role in delaying the time to progression; however, there was no statistically significant relationship between OS and treatment modality.

## 1. Introduction

Lung cancer remains the leading cause of cancer-related mortality worldwide, with non-small cell lung cancer (NSCLC) being the most prevalent subtype. Approximately 10–20% of NSCLC patients present with brain metastases at the time of diagnosis, with lung cancer accounting for 40–50% of all cases of brain metastases [1,2]. These brain metastases contribute substantially to morbidity and mortality, presenting significant challenges in management due to their complex biology and impact on neurological function. 

The advent of molecular profiling has revolutionized the treatment landscape for lung cancer, particularly in cases harboring targetable mutations such as *EGFR*, *ALK*, *ROS1*, and, more recently, *KRAS*. These mutations have facilitated the development and implementation of targeted systemic therapies, which have demonstrated efficacy in controlling extracranial disease [3,4]. Notably, trials like FLAURA and AURA3 have demonstrated that osimertinib, when used in *EGFR* mutation-positive metastatic NSCLC with CNS involvement, significantly improves progression-free survival (PFS) and overall survival (OS) [5,6,7,8]. 

Despite the advancements in systemic therapies and radiotherapy techniques, the optimal management of brain metastases following surgical resection remains a subject of ongoing debate. Surgical resection is often pursued in patients with solitary or symptomatic brain metastases, offering the immediate relief of the mass effect and the potential for improved survival. Following resection, radiation therapy, including whole-brain radiation therapy (WBRT) and stereotactic radiosurgery (SRS), has been a cornerstone in managing brain metastases [9]. In the post-operative setting, SRS has been shown to significantly improve local control, as demonstrated in a phase III clinical trial by Mahajan et al., which reported reduced local recurrence rates at 12 months [10].

The integration of targeted therapies and immunotherapies capable of penetrating the blood–brain barrier (BBB) has further revolutionized the treatment options for patients with actionable mutations [3,4]. Consequently, the treatment paradigms for brain metastases have evolved into a trimodality approach, incorporating surgical resection, radiation therapy (SRS/WBRT), and targeted systemic agents. While the evidence supports that combining targeted therapies with radiation therapy enhances intracranial tumor control compared to standalone systemic therapy, the rapid development of efficacious immunotherapies has introduced a debate on whether radiation therapy might be deferred in some cases [11,12,13]. 

With a push towards the molecular characterization of cancers and systemic therapy advancements, we intended to examine outcomes by treatment modality at our institution with respect to prioritizing RT and systemic therapies following the surgical resection of brain metastases. This study sought to investigate whether the inclusion of intracranial radiotherapy and/or systemic therapy offered benefits for NSCLC patients with brain metastases.

## 2. Materials and Methods

Following Institutional Review Board approval, the patient data were collected from the electronic medical records across multiple sites at a single institution. This study included patients with NSCLC who developed brain metastases and underwent surgical resection between 2014 and 2023. Patients were included if they received upfront surgical resection followed by adjuvant therapy i.e., radiation and/or systemic therapy. Patients were excluded if they lacked clinical follow-up and post-treatment radiologic imaging and did not receive adjuvant therapy.

The patients were divided into cohorts based on treatment: (1) radiation therapy (RT) alone, (2) systemic therapy alone, and (3) combined therapy (RT + systemic therapy). For the patients who received RT, the treatment was administered using Gamma Knife SRS, LINAC-based SRS, or WBRT. The SRS dosing ranged from 18–20 Gy in a single fraction to 24–27 Gy in three fractions. For the patients who received WBRT, the dose ranged from 20 to 30 Gy in 5–10 fractions. The systemic therapy treatment options typically included platinum-based chemotherapy, immunotherapy, and targeted therapy such as first- to third-generation tyrosine kinase inhibitors (TKIs).

The baseline patient characteristics including age, gender, race, ethnicity, smoking history, pathology, number of brain metastases, presence of leptomeningeal disease, RT dose/fractionation, and types of systemic therapy were collected. The primary endpoints were in-brain freedom from progression (FFP) and overall survival (OS). In-brain FFP included any intraparenchymal metastatic brain lesion. The secondary endpoint was index lesion recurrence—with the index lesion being defined as the metastasis that was surgically resected. Toxicity following RT was included, specifically the rate of radiation necrosis based on a post-treatment MRI. The patients were evaluated based on their treatment modality: surgery with RT (SRS and WBRT), surgery with systemic therapy (TKIs, immunotherapy, and chemotherapy), or surgery with combination therapy. The tumor molecular profiles were collected, including *PD-L1*, *EGFR*, *ALK fusion*, *KRAS*, *RET fusion*, *NTRK fusion*, *ROS1 fusion*, *MET Ex 14 skipping*, *HER2 mutation*, *BRAF V600E*, *TP53*, and *STK11* mutations. The dataset is provided in the Appendix A.

### Statistical Analysis

The in-brain FFP and OS were analyzed using Kaplan–Meier (KM) curves, with the treatment groups compared using the log-rank test. The index lesion recurrence based on treatment modality was analyzed using Fisher’s exact test and the chi-squared (χ^2^) test. The categorical variables between the patient treatment cohorts were compared using the chi-squared test, and the continuous variables were compared using independent two-sample t-tests. A multivariate regression analysis was also performed to adjust for the baseline patient characteristics when looking at the in-brain FFP and OS. The primary and secondary endpoints were also analyzed with respect to the molecular data and specific mutations. The number of patients for which molecular data were collected between 2008 and 2015 was compared to those between 2016 and 2022 using the χ^2^ test. A *p* value of <0.05 was considered statistically significant, and all tests were two-tailed. All the statistical analyses were performed using SAS Studio version 4.4.

## 3. Results

### 3.1. Patient Characteristics

A total of 272 patients were identified, with 162 (60%) having adequate follow-up data, with a median follow-up duration of 20 months. Seven of the one hundred and sixty-two patients who did not receive adjuvant therapy after their surgery were excluded. The median age at the diagnosis of the brain metastasis was 65 years old (range = 33–83), with the majority of patients being women (59.3%) and White/non-Hispanic (65.2%). All the patients were either RPA class 1 or class 2. Following the surgical resection of the index brain lesion, 65.8% (102/155) of the patients received adjuvant combination radiotherapy and systemic therapy, while 34.2% (53/155) received adjuvant monotherapy. There was a total of 148 patients who received RT, of which 13 received WBRT, 59 received single fraction SRS (sfSRS), and 74 received fractionated SRS (fSRS). There were two patients who were missing the RT dose and fractionation documentation. WBRT was often recommended in patients with numerous brain metastases. The choice of sfSRS or fSRS was decided based on the post-operative tumor bed cavity size and physician discretion. Twelve (7.7%) patients experienced radiation necrosis following RT. The patients who received chemotherapy either had platinum-based chemo or pemetrexed alone. With respect to immunotherapy and targeted therapy, forty-seven patients received pembrolizumab, nineteen patients received TKIs (i.e., alectinib, erlotinib, crizotinib, osimertinib, and dabrafenib), two patients received nivolumab, two patients received ipilumimab with nivolumab, two patients received atezolizumab, and one patient received durvalumab. The baseline characteristics are summarized in Table 1.

### 3.2. In-Brain Freedom from Progression Analysis

The median freedom from progression (FFP) time for the whole patient cohort was 9.7 months. The patients who were treated with a combination of adjuvant RT and systemic therapy had a median FFP time of 10.72 months (IQR: 5.71–25.51 months), compared to 5.38 months (IQR: 2.75–18.92 months) for those who received adjuvant monotherapy (log-rank test, *p* = 0.072). The Kaplan–Meier (KM) curve is displayed in Figure 1. 

In total, 148 patients received RT in the patient cohort. Out of these 148 patients, the patients receiving systemic therapy had a median FFP time of 10.72 months (IQR: 5.71–25.51 months), as opposed to 5.13 months (IQR: 2.30–18.20 months) without systemic therapy (log-rank test, *p* = 0.083) (Figure 2). 

Amongst the 109 patients who received systemic therapy, there was no significant difference observed in the FFP between the patients receiving RT (median FFP: 10.72 months, IQR: 5.71–25.51 months) and those not receiving RT (median FFP: 12.13 months, IQR: 3.44–22.39 months) (*p* = 0.409) (Figure 3).

When stratifying patients by treatment modality, there was no significant difference between the patients receiving combination therapy vs. RT alone vs. systemic therapy alone (Figure 4). The median FFP times for the patients who received adjuvant RT, systemic therapy, or combined therapy were 5.13 months, 12.13 months, and 10.72 months, respectively. Additionally, a multivariate regression analysis did not show any other factors associated with in-brain FFP.

### 3.3. Overall Survival and Index Lesion Recurrence Analysis

Regarding overall survival (OS), there was no significant difference found between the patients receiving combined therapy (median OS: 47.48 months) and those receiving a single modality therapy (median OS: 53.18 months) (Figure 5, *p* = 0.982). The monotherapy group was composed of forty-six patients who received RT alone and seven patients who received systemic therapy alone. The Kaplan–Meier curve showed no significant difference in the median OS observed when patients were stratified by adjuvant treatment modality (Figure 6, *p* = 0.839).

Adjusting for the baseline characteristics, the multivariate regression analysis showed that the patients who were of Hispanic/Latino descent had a statistically significant worse OS rate compared to patients who were non-Hispanic/Latino with a median OS of 12.75 months vs. 53.15 months, respectively (*p* = 0.015). The Kaplan–Meier curve is shown in Figure 7.

The index lesion recurrence was analyzed based on treatment modality: RT, systemic therapy, immunotherapy, and combination therapy. There was no association with preventing an index lesion recurrence in the patients treated with RT (*p* = 0.241) and systemic therapy (*p* = 0.122). However, there was a positive trend towards a reduction in index lesion recurrences in the patients who received multimodality therapy (*p* = 0.063). Furthermore, there was a statistically significant reduction in index lesion recurrences when looking at the patients who received immunotherapy (*p* = 0.0018). This immunotherapy analysis was broken down into patients receiving RT, chemotherapy, and no chemotherapy with *p*-values of 0.0021, 0.0066, and 0.067, respectively.

### 3.4. Molecular Analyses

The distribution of patients with actionable mutations is shown in Figure 8. Testing for the molecular profiles of the tumors significantly increased between 2016 and 2022 compared to the previous years (2008–2015) (χ^2^ test, *p* < 0.0001). With respect to in-brain FFP and OS, there was no association between the mutational status and outcomes.

## 4. Discussion

This study provides valuable insights into the treatment outcomes of patients with brain metastases from non-small cell lung cancer (NSCLC) who underwent surgical resection followed by various adjuvant therapies including RT and systemic therapy (immunotherapy/TKIs and/or chemotherapy). Our findings emphasize the critical role of combining radiotherapy and systemic therapy in improving intracranial disease control, particularly in delaying in-brain progression and an index lesion recurrence following the surgical resection.

One notable observation is the trend toward improved in-brain FFP in the patients who received systemic therapy. The patients treated with radiation therapy with and without systemic therapy had a median FFP of 10.72 months compared to 5.13 months, respectively (*p* = 0.083). This suggests that systemic therapy, including targeted therapies and immunotherapy, plays a crucial role in delaying intracranial progression when given in conjunction with RT. The benefit of combination therapy was also evident, as the patients who received both had a median FFP of 10.72 months compared to 5.38 months for the patients receiving monotherapy (*p* = 0.072). These results align with previous studies indicating that a multimodal approach enhances intracranial tumor control compared to single modality treatments [14,15]. Additionally, in-brain FFP was not statistically significant with respect to RT, as a majority of the patient cohort received single fraction or fractionated SRS (13 patients received WBRT). This indicates that systemic therapy should be prioritized when attempting to control for brain metastases outside of the surgical tumor bed [16]. 

Our study did not demonstrate a significant difference in OS between the various treatment modalities. The patients receiving combined therapy had a median OS of 47.48 months and those receiving a single modality therapy had a median OS of 53.18 months (*p* = 0.982). This finding could be due to the multifactorial nature of survival in NSCLC patients, where extracranial disease progression plays a significant role [17]. Additionally, many other clinical factors can affect overall survival, such as age, performance status, and gender [18]. The lack of a significant difference in OS highlights the complexity of treating metastatic NSCLC and suggests that while intracranial control is crucial, systemic disease management remains equally important. As systemic therapy options continue to improve, particularly with advancements in the understanding of driver mutations, the optimal treatment approach is likely to evolve [19,20]. At our institution, as a standard policy, we address the post-operative cavity 14 days after surgery with radiotherapy (pending surgical clearance) based on the outcomes data on the local control. Systemic therapy follows the completion of adjuvant radiation. The absence of a significant OS difference underscores the need for ongoing research to optimize systemic therapies and their integration with local treatments to improve overall outcomes in this patient population.

When we adjusted for the baseline characteristics, the multivariate regression analysis revealed that the patients of Hispanic/Latino descent had a statistically significant worse OS of 12.75 months when compared to the overall survival of the remaining patient cohort. There are limited data in the literature regarding Hispanic/Latino patients with NSCLC and brain metastases; however, a study utilizing the SEER database demonstrated that Hispanic patients were less likely to receive radiation and less likely to use supportive medication such as opioids, antiepileptics, steroids, and antidepressants [21]. Hispanic or Latino patients often have a higher likelihood of presenting with advanced NSCLC and comorbidities that can complicate the treatment [22]. Socioeconomic barriers, language differences, and potential disparities in healthcare access can also impact the diagnosis, timely treatment, and access to newer targeted therapies [23]. Ultimately, for Hispanic/Latino patients, improving access to advanced care and addressing potential genetic predispositions or mutations are critical to optimizing outcomes. Ethnicity-specific studies are needed to better understand and address these disparities in lung cancer care.

Our results also indicate a positive trend towards reduced index lesion recurrences with the use of systemic therapy combined with RT (*p* = 0.063). When systemic therapy or radiotherapy is given alone, there was no statistically significant effect on index lesion recurrences. However, there are phase III studies that demonstrate the efficacy of post-operative RT as detailed in Mahajan et al. [10]. A large limitation to this analysis is the imbalance of patients with respect to RT. Only seven patients in our cohort did not receive RT. All the patients were discussed in our CNS multidisciplinary tumor board where the recommendations were based on level 1 evidence or the best available evidence. All the patients were referred for radiotherapy consultation as per the standard of care; however, there were several factors that interfered, such as an inability to tolerate mask immobilization, a poor KPS, the patient’s decision against RT, being lost to follow-up, and post-operative complications. Furthermore, index lesion recurrences were statistically significantly reduced in patients who received immunotherapy in combination with RT (*p* = 0.0021). This suggests a potential synergistic effect when combining radiation therapy with targeted systemic treatments. The literature supports this observation, particularly in studies utilizing SRS with TKIs with BBB penetration to improve survival and intracranial control [24,25,26,27]. The exact mechanism through which synergism occurs is unclear, but the elusive abscopal effect has been reported to be a potential theory [28,29].

Testing for molecular markers has increased significantly in recent years and has become part of the management of patients with lung cancers. There was a significant difference between the number of patients getting tested for molecular sequencing during 2008–2015 compared to 2016–2022 (*p* < 0.0001). The integration of molecular characterization into the treatment planning process is essential, as evidenced by the distinct outcomes associated with specific mutations [30,31]. The identification of actionable mutations, such as *EGFR*, *ALK*, *PD-L1*, and *ROS-1*, has transformed the therapeutic landscape, allowing for more personalized and effective treatment strategies [32,33,34,35]. The recently published TURBO trial demonstrated an improved time to CNS progression and local CNS control when TKIs were combined with SRS [36]. Although our study did not show an improvement in OS when stratifying by treatment modality, this may be attributed to the imbalance of patients with and without actionable mutations in our study. In this study, we did not find a statistically significant relationship between certain mutations and patient outcomes; however, the limited number of patients with actionable mutations in this study highlights the need for larger cohorts to validate these findings and fully understand the implications of different genetic profiles.

## 5. Conclusions

Our study underscores the critical role of systemic therapy, particularly when combined with radiotherapy, in improving intracranial control in patients with brain metastases from non-small cell lung cancer. This combination showed that a positive trend in the reduction in in-brain freedom from progression (FFP) and immunotherapy was shown to reduce index lesion recurrences when combined with RT. These findings highlight the value of a multidisciplinary approach. Although the treatments did not significantly impact overall survival, these findings emphasize the ongoing need for comprehensive strategies that address both intracranial and systemic diseases. The importance of molecular profiling in guiding treatment decisions cannot be overstated, as specific mutations significantly influence patient outcomes. 

Future research should focus on larger, prospective studies to validate these findings and explore the integration of novel systemic therapies with radiotherapy, aiming to enhance the survival of and quality of life for patients with metastatic NSCLC. Future research can elucidate the optimal integration with radiation therapy to maximize both intracranial and systemic disease control. Additionally, the significant disparities in survival outcomes based on ethnicity highlight the need for more focused studies on genetic predispositions and socioeconomic factors that may affect access to care and treatment responses. Larger, prospective studies with more comprehensive documentation of treatment timing, side effects, and molecular profiling are essential to validate these findings and refine personalized treatment approaches.

## Figures and Tables

**Figure 1 cancers-16-03270-f001:**
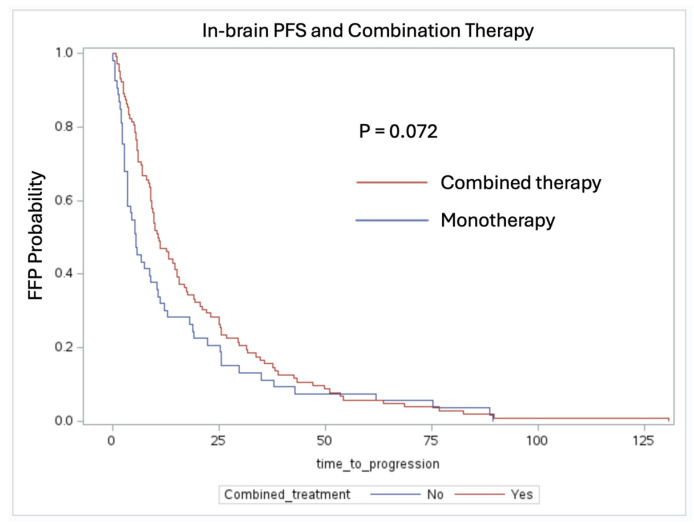
Kaplan–Meier analysis of in-brain FFP stratified by combination therapy including adjuvant RT + systemic therapy vs. adjuvant monotherapy (log-rank test, *p* = 0.072). Patients treated with a combination of RT and systemic therapy vs. adjuvant monotherapy had a median FFP time of 10.72 months vs. 5.38 months, respectively.

**Figure 2 cancers-16-03270-f002:**
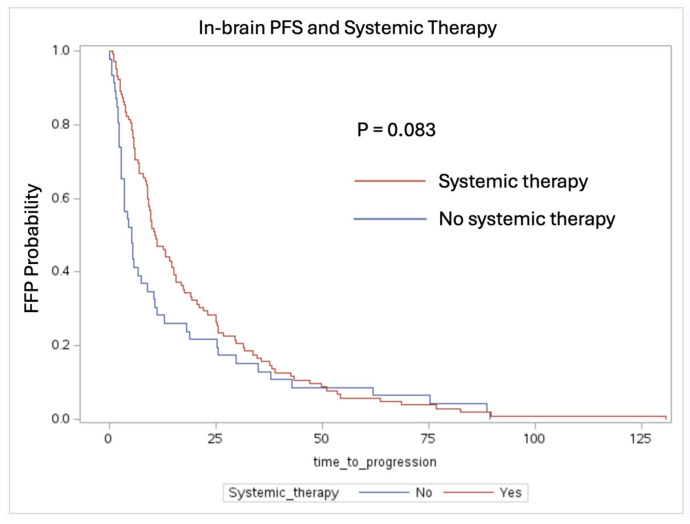
In-brain FFP stratified by patients receiving adjuvant systemic therapy (log-rank test, *p* = 0.083). In patients who received RT, treatment with systemic therapy was associated with a median FFP of 10.72 months vs. 5.13 months without systemic therapy.

**Figure 3 cancers-16-03270-f003:**
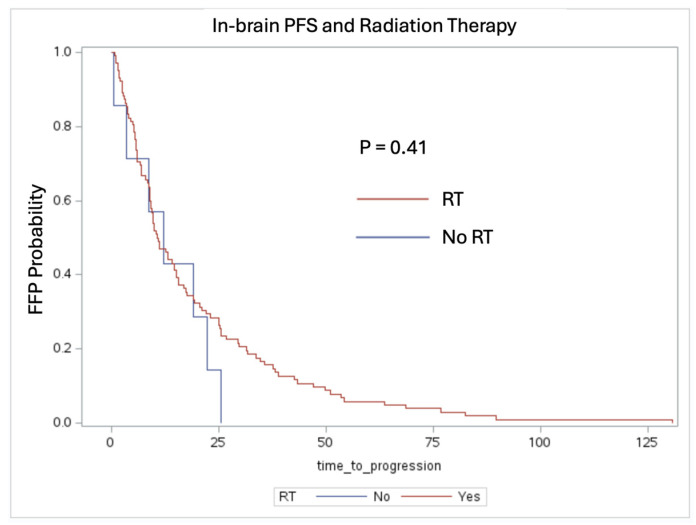
In-brain FFP stratified by patients receiving systemic therapy with and without RT (log-rank test, *p* = 0.41). Patients treated with RT vs. no RT had a median FFP time of 10.72 months vs. 12.13 months, respectively.

**Figure 4 cancers-16-03270-f004:**
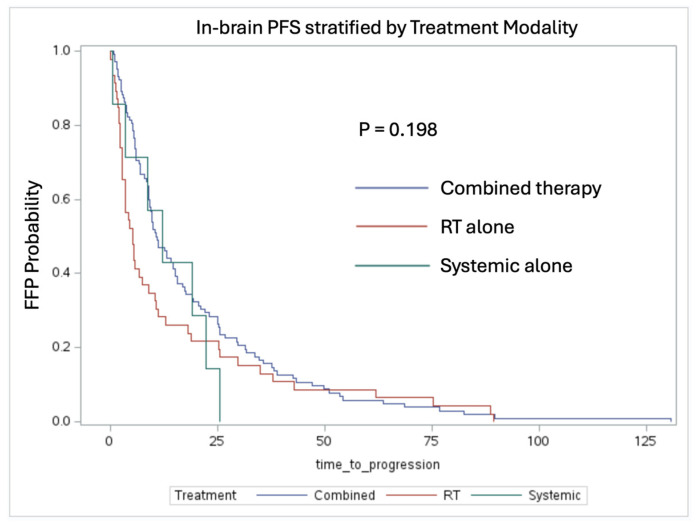
In-brain FFP stratified by treatment modality (log-rank test, *p* = 0.227). Patients who received adjuvant RT alone, systemic therapy alone, or combined therapy had a median FFP time of 5.13 months, 12.13 months, and 10.72 months, respectively.

**Figure 5 cancers-16-03270-f005:**
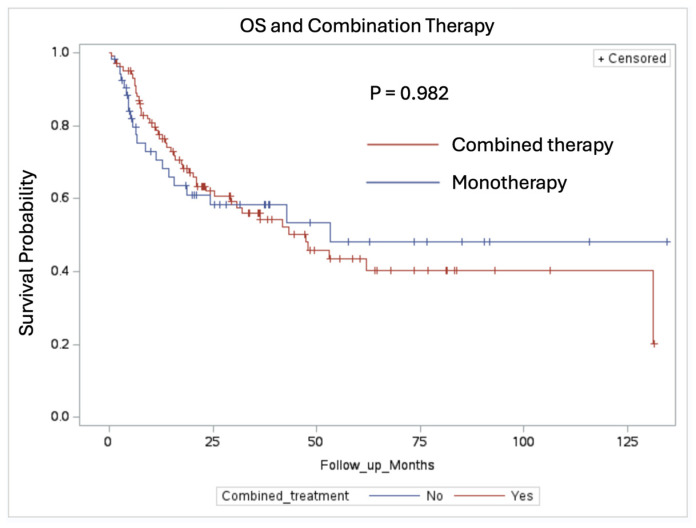
Comparing OS in patients receiving combined therapy vs. single modality therapy, *p* = 0.922. Patients who received combined therapy had a median OS of 47.48 months and patients who received monotherapy had a median OS of 53.18 months.

**Figure 6 cancers-16-03270-f006:**
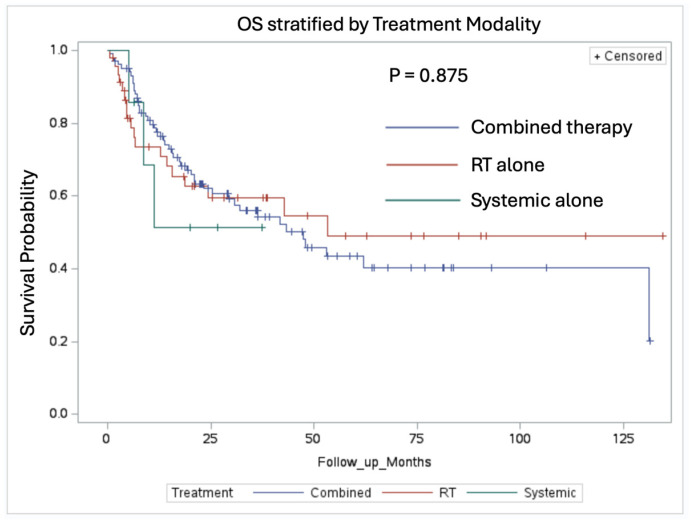
Comparing OS in patients stratified by treatment modality. There was no significant difference in OS, *p*-value = 0.875.

**Figure 7 cancers-16-03270-f007:**
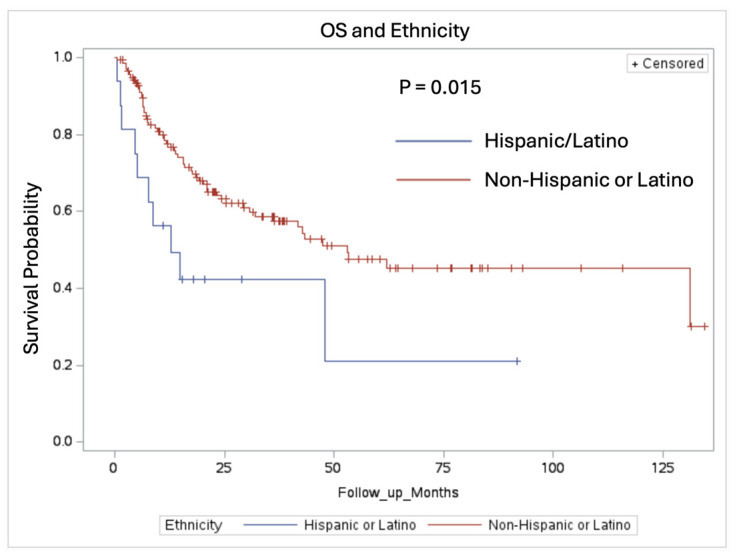
Overall survival in patients stratified by ethnicity. Median OS for patients of Hispanic or Latino descent vs. non-Hispanic/Latino descent were 12.75 months vs. 53.15 months (*p* = 0.015).

**Figure 8 cancers-16-03270-f008:**
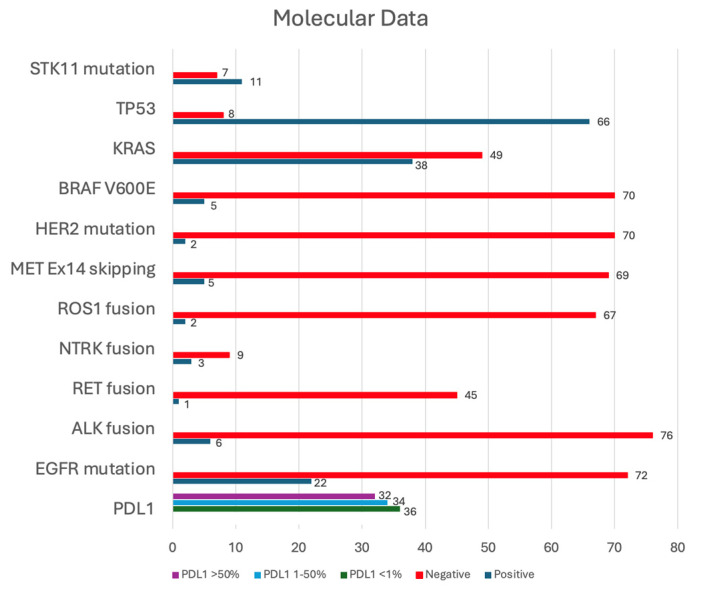
Number of patients tested for actionable mutations including an *STK11 mutation*, *TP53*, *KRAS*, *BRAF V600E*, *HER2*, *MET Ex14 skipping*, *ROS1 fusion*, *NTRK fusion*, *RET fusion*, *ALK fusion*, *EGFR mutation*, and *PD-L1*.

**Table 1 cancers-16-03270-t001:** Patient baseline characteristics including demographics and clinical characteristics.

Characteristics	Patients (*n* = 155)
Age at brain metastasis diagnosis, years Median (range)	65 (33–83)
Gender, *n* (%)	
Male Female	63 (40.7) 92 (59.3)
Race, *n* (%)	
Asian	13 (8.4)
Black	25 (16.1)
White/Hispanic	16 (10.3)
White/non-Hispanic	101 (65.2)
Smoking status, *n* (%)	
Never	27 (17.4)
Former	85 (54.8)
Current	43 (27.8)
NSCLC histology, *n* (%)	
Adenocarcinoma	126 (81.3)
Squamous cell carcinoma	21 (13.6)
NSCLC NOS/poorly differentiated	8 (5.1)
Number of brain metastases	
1–4	136 (87.8)
5–10	16 (10.3)
>10	3 (1.9)
Leptomeningeal disease	4 (2.6)
Adjuvant therapy, *n* (%)	
Systemic therapy only	7 (4.5)
Radiation therapy only	46 (29.7)
Combined modality therapy	102 (65.8)
Systemic therapy, *n* (%)	
Chemotherapy alone	59 (38.1)
Immunotherapy alone	73 (47.1)
Chemo and immuno	30 (19.4)
Toxicity, *n* (%)	
Radiation necrosis	12 (7.7)

## Data Availability

The raw data supporting the conclusions of this article will be made available by the authors on request.

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
