# Peer review of "Prioritizing Radiation and Targeted Systemic Therapies in Patients with Resected Brain Metastases from Lung Cancer Primaries with Targetable Mutations: A Report from a Multi-Site Single Institution"

_cancers, 2024, doi:10.3390/cancers16193270_

Round 1
Reviewer 1 Report
Comments and Suggestions for Authors
1. Please add annotation of different colored lines and p values between each group in the fig instead of in the fig legend.
2. In fig 7, are those genetic alterations associated with patients’ survival?
3. Multivariant analyses should be done to see if any clinical factors are associated with patients’ survival.
Comments on the Quality of English Languagewell written
Author Response
Comment 1: Please add annotation of different colored lines and p values between each group in the fig instead of in the fig legend.
Response 1: The line annotations and the p-values of each figure have been added to each figure.
Comment 2: In fig 7, are those genetic alterations associated with patients’ survival?
Response 2: The genetic alterations were no associated with patients' survival including both in-brain PFS and OS. I have added a comment on this in the results section 3.4.
Comment 3: Multivariant analyses should be done to see if any clinical factors are associated with patients’ survival.
Response 3: Thank you for this comment. I have added a multivariate analysis to account for baseline characteristics and there was a significant finding of OS being affected by ethnicity, Hispanic/Latino vs non-Hispanic/Latino. This has been added in section 3.3, lines 199-202.
Reviewer 2 Report
Comments and Suggestions for Authors
This manuscript can not be considered a review, just a part of a proper review.
More references, more results. It must be MORE!
Too many spaces between lines.
Author Response
Comment 1: This manuscript cannot be considered a review, just a part of a proper review.
Response 1: You are correct: this article is not meant to be a comprehensive review article of the management of NSCLC brain mets. This is a retrospective study of patients with brain metastases managed at our institution. The literature has been referenced in the context of our data. I have added additional results and findings from our study.
Comment 2: More references, more results. It must be MORE!
Response 2: Further analyses have been performed including multivariate regression analyses in terms of OS and PFS, chi-square and Fisher's exact test for index lesion recurrence with respect to multiple treatment paradigms specifically immunotherapy which was statistically significant. We have included results of the MVA and found that ethnicity (Hispanic/Latino vs non-Hispanic/Latino) had an effect on OS. Results regarding OS and PFS with respect to molecular data is provided. Additional baseline characteristics have been included. Patients with radiation necrosis were included. The number of patients treated with WBRT vs single-fraction SRS vs fractionated SRS were included. The types of chemotherapy and immunotherapy along with the number of patients who received it were included.
Comment 3: Too many spaces between lines.
Response 3: Spacing has been adjusted and checked.
Reviewer 3 Report
Comments and Suggestions for Authors
The authors report the results of a single institution retrospective study evaluating adjuvant therapy for brain metastases following surgery.
The background and rationale should be clarified. It is generally accepted that the standard of care for brain metastases following resection is post-operative radiation therapy. Whether via whole brain radiotherapy, or radiosurgery, this approach is validated in phase 3 randomized trials. The ideal comparison should the current standard of surgery and post-operative radiation therapy versus surgery, post-operative radiation therapy and systemic therapy.
Patients who receive surgery alone should be excluded as this is not standard of care.
The multidisciplinary rationale determining how the patient's treatment plan was determined should be described. Was it tumor board standard, existing clinical trials, physician preference leading to adjuvant treatment selection?
The timing of surgery to adjuvant radiation therapy, adjuvant systemic therapy and duration of therapies should be discussed. There are multiple studies on treatment package time, and timing of radiation and systemic therapy, on local control.
The selection of whole brain therapy versus radiosurgery versus fractionated radiosurgery should be detailed.
Presence of leptomeningeal disease should be included in patient demographics.
Extracranial disease burden (oligometastatic or polymetastatic) and whether the primary site is controlled should be included.
Number and types of prior systemic therapies and prior radiation therapy, especially intracranial radiation therapy, should be included in the data.
Whether patients presented with de novo metastatic disease or progressed to metastatic disease should be included in the data
Side effects / toxicities (radiation necrosis in particular) should be included in the results
Comments on the Quality of English LanguageThe are several minor grammatical errors in the manuscript.
eg: line 27 ".., advances molecular in..."
Many of the sentences can be written in a more concise manner: 3g: line 29 ..."we sought out to examine" can be rewritten as "we evaluated" or "we reviewed"
Author Response
Comment 1: The background and rationale should be clarified. It is generally accepted that the standard of care for brain metastases following resection is post-operative radiation therapy. Whether via whole brain radiotherapy, or radiosurgery, this approach is validated in phase 3 randomized trials. The ideal comparison should the current standard of surgery and post-operative radiation therapy versus surgery, post-operative radiation therapy and systemic therapy.
Response 1: Agreed that the standard of care for brain metastases is post-operative RT. I have removed the patients who did not receive adjuvant therapy and only included patients who have gotten adjuvant therapy, either RT and/or systemic therapy.
Comment 2: Patients who receive surgery alone should be excluded as this is not standard of care.
Reponse 2: Patients who did not receive adjuvant therapy have been removed and all analyses have been repeated. All data in the manuscript has been updated.
Comment 3: The multidisciplinary rationale determining how the patient's treatment plan was determined should be described. Was it tumor board standard, existing clinical trials, physician preference leading to adjuvant treatment selection?
Response 3: All patients are discussed in our CNS multidisciplinary tumor board, where the recommendations are being made based on level 1 or best available evidence (this has been added to the manuscript). The standard of care was offered for every patient. All of the patients presented in this manuscript were offered RT after surgery. Only 7 patients did not receive RT for specific reasons that I have added into the manuscript, lines 280-285. Treatment options are based on all of the above that you have mentioned.
Comment 4: The timing of surgery to adjuvant radiation therapy, adjuvant systemic therapy and duration of therapies should be discussed. There are multiple studies on treatment package time, and timing of radiation and systemic therapy, on local control.
Response 4: At our institution, as a standard policy, based on the outcomes data on local control, we typically address post-operative cavity 14 days after the surgery with radiotherapy (pending surgical clearance). Systemic therapy follows the completion of adjuvant radiation. This has been added to the manuscript in lines 256-259
Comment 5: The selection of whole brain therapy versus radiosurgery versus fractionated radiosurgery should be detailed.
Response 5: Thank you for this comment. In our cohort, only 13 patients received adjuvant WBRT. WBRT was likely chosen based on the extensive number of brain metastases or concern for leptomeningeal disease. Single fraction vs fractionated SRS was likely based on post-op cavity size and at the treating physician's clinical evaluation. This is included in lines 136-138.
Comment 6: Presence of leptomeningeal disease should be included in patient demographics.
Response 6: Out of the 155 patients, 4 patients were documented to have concern for leptomeningeal disease at diagnosis. I have included this in the patient demographics.
Comment 7: Extracranial disease burden (oligometastatic or polymetastatic) and whether the primary site is controlled should be included.
Response 7: All patients to our knowledge were RPA class I or RPA class II which I have included in lines 130-131.
Comment 8: Number and types of prior systemic therapies and prior radiation therapy, especially intracranial radiation therapy, should be included in the data.
Response 8: I have included the types of systemic therapies that these patients received. Lines 137-141. Majority of patients that were chemotherapy candidates received platinum-based chemotherapy or pemetrexed. The breakdown of the number of patients receiving chemotherapy vs immunotherapy vs chemoimmunotherapy is added to the baseline characteristics table.
Comment 9: Whether patients presented with de novo metastatic disease or progressed to metastatic disease should be included in the data
Response 9: Our patients largely were patients with known lung cancer which progressed to metastatic disease in the brain.
Comment 10: Side effects / toxicities (radiation necrosis in particular) should be included in the results
Response 10: I have included this data into the clinical characteristics table: 12 patients (7.7%) of patients who received RT had radiation necrosis based on MRI findings.
Round 2
Reviewer 3 Report
Comments and Suggestions for Authors
In the contemporary era, and in ongoing phase II and III trials, patients with brain metastases from lung cancer are being simultaneously stratified by performance status, molecular status, intracranial and extracranial disease burden, systemic therapy and radiation therapy modality (please see active trials on clinicaltrials.gov for reference). The patient cohort in this study should be evaluated at this level of granularity to sufficiently support conclusions relevant to the multidisciplinary management of lung cancer brain metastases. However, there is a high probability of lacking sufficient power to run analyses on these stratified subsets. The study may need to narrow its focus to subsets of patients with sufficient numbers for analyses or consider pooling data from multiple institutions.
Comments on the Quality of English LanguageEnglish is overall acceptable